# Methods of Sandy Land Detection in a Sparse-Vegetation Scene Based on the Fusion of HJ-2A Hyperspectral and GF-3 SAR Data

Yi Li [1], Junjun Wu [2,*], Bo Zhong [2], Xiaoliang Shi [1], Kunpeng Xu [3], Kai Ao [2], Bin Sun [3], Xiangyuan Ding [3], Xinshuang Wang [4], Qinhuo Liu [2], Aixia Yang [2], Fei Chen [1] and Mengqi Shi [1]

1. School of Surveying and Mapping, Xi'an University of Science and Technology, Xi'an 710054, China; 20210061035@stu.xust.edu.cn (Y.L.); xiaoliangshi@xust.edu.cn (X.S.); 20210010002@stu.xust.edu.cn (F.C.); 20210061026@stu.xust.edu.cn (M.S.)
2. State Key Laboratory of Remote Sensing Science, Aerospace Information Research Institute, Chinese Academy of Sciences, Beijing 100101, China; zhongbo@aircas.ac.cn (B.Z.); aokai@aircas.ac.cn (K.A.); liuqh@aircas.ac.cn (Q.L.); yangax@aircas.ac.cn (A.Y.)
3. Institute of Forest Resource Information Techniques, Chinese Academy of Forestry, Beijing 100091, China; xukp@ifrit.ac.cn (K.X.); sunbin@ifrit.ac.cn (B.S.); dingxiangyuan@ifrit.ac.cn (X.D.)
4. Shaanxi Basic Geographic Information Center, Ministry of Natural Resources, Xi'an 710054, China; w.xinshuang@gmail.com
* Correspondence: wujj@aircas.ac.cn

**Abstract:** Accurate identification of sandy land plays an important role in sandy land prevention and control. It is difficult to identify the nature of sandy land due to vegetation covering the soil in the sandy area. Therefore, HJ-2A hyperspectral data and GF-3 Synthetic Aperture Radar (SAR) data were used as the main data sources in this article. The advantages of the spectral characteristics of a hyperspectral image and the penetration characteristics of SAR data were used synthetically to carry out mixed-pixel decomposition in the "horizontal" direction and polarization decomposition in the "vertical" direction. The results showed that in the study area of the Otingdag Sandy Land, in China, the accuracy of sandy land detection based on feature-level fusion and single GF-3 data was verified to be 92% in both cases by field data; the accuracy of sandy land detection based on feature-level fusion was verified to be 88.74% by the data collected from Google high-resolution imagery, which was higher than that based on single HJ-2A (74.17%) and single GF-3 data (88.08%). To further verify the universality of the feature-level fusion method for sandy land detection, Alxa sandy land was also used as a verification area and the accuracy of sandy land detection was verified to be as high as 88.74%. The method proposed in this paper made full use of the horizontal and vertical structural information of remote sensing data. The problem of mixed pixels in sparse-vegetation scenes in the horizontal direction and the problem of vegetation covering sandy soil in the vertical direction were both well solved. Accurate identification of sandy land can be realized effectively, which can provide technical support for sandy land prevention and control.

**Keywords:** sandy land; mixed pixel decomposition; polarization decomposition; support vector machine classification; image fusion

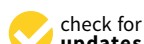



## 1. Introduction

In 2015, the United Nations published Transforming Our World: The 2030 Agenda for Sustainable Development Goals (SDGs). Therein, SDG 15 states explicitly to, "By 2030, combat desertification, restore degraded land and soil, including land affected by desertification, drought and floods, and strive to achieve a land degradation-neutral world". The expansion of sandy land not only poses a serious threat to the sustainable development of human life [1–3] but also is a global problem related to biodiversity loss, deforestation, and soil degradation [4–6]. Desertification refers to the degradation process of the natural environment in arid, semi-arid, and even sub-humid areas due to the combined effects of

human activities and climate change [7]. The dominant direct physical processes responsible for desertification are water erosion, wind erosion, and salinization [8]. Strong winds, insufficient soil and air humidity, scarcity of water and vegetation cover, drought, and soil erosion are the main characteristics of the desert [9,10]. Desertification in the arid and semi-arid areas of northern China is one of the most typical ecological and environmental problems, and it is an important content to be focused on and managed [11–13]. Accurately identifying sandy soil and in real time detecting the distribution of sand land is an effective way to prevent desertification for regional ecological and environmental protection and sustainable development [14,15]. Under long-term control and protection, the area of sand land is shrinking and desertification has been curbed as a whole [16,17]. However, there is still degeneration in some areas, and technology supports must be improved to implement desert prevention and control projects.

According to different vegetation coverage, sandy land can be divided into fixed sandy land, semi-fixed sandy land, and shifting sandy land. The characteristics of shifting sandy land are obvious and easy to identify, but the influence of vegetation cover makes it more difficult to identify the sandy soil in fixed sandy land and semi-fixed sandy land. Remote sensing technology has provided a more objective and accurate data basis for the monitoring and evaluation of sandy land due to its wide observation range, the fact that it provides real-time information, and its dynamics [18–21], and it has become one of the indispensable methods of monitoring sandy land on a regional and even global scale [22]. A variety of remote sensing methods of monitoring sandy land have been proposed by different scholars. Quantitative inversion based on soil characteristic parameters [23–27] and mixed-pixel decomposition [28] were used as important methods of sandy land detection. In the research on sandy land detection based on quantitative inversion of soil characteristic parameters, the measured data on soil characteristic parameters obtained by investigation or analysis in a small area was used for remote sensing quantitative inversion modeling and verification and then used for large-scale sandy land detection [29]. Mixed-pixel decomposition assumes that each pixel is composed of several "pure" endmembers and pixel spectral can be decomposed into the proportion of several endmembers that contribute to the pixel signal [30]. Generally speaking, mixed-pixel decomposition has been mainly applied to hyperspectral data [31–33], and it is rarely applied to multispectral data because of the limited bands. In the research on sand detection based on mixed-pixel decomposition, most research has just identified the abundance of sandy land [34], which still has a certain deviation from real sandy land. In terms of the above problems, accurate identification of sandy soil was achieved by determining a reasonable threshold and its accuracy had reached more than 80% [28]. However, the problem of vegetation covering the nature of sandy land has not been well resolved. Therefore, detecting the essence of sand land and penetrating vegetation cover by remote sensing technology are problems that need to be urgently solved.

SAR data can be obtained clearly under all weather conditions and has a penetration characteristic in the vertical direction. Polarization decomposition technology is a new method developed in the last two decades to reveal the scattering mechanism of ground objects [35,36]. However, this method is mainly used in qualitative research for remote sensing image classification. Accurate identification of sandy land can be realized by using the main scattering features of ground objects that have been effectively extracted by polarization decomposition methods [37]. Hyperspectral remote sensing data have the advantage of high spectral resolution, and sensitive bands can be captured due to its continuous spectral of ground features, which greatly improves the ability to detect sandy land. Generally speaking, remote sensing fusion technology can be divided into three different levels: pixel level, feature level, and decision level. Pixel-level fusion is to fuse multiple data sources with a single resolution, which was used to extract and classify the land cover [38]. Feature-level fusion refers to extracting features from different data sources and then merging them into one or more feature maps to replace the original data. Feature-level fusion and decision-level fusion were used to achieve land cover classification in

urban areas [39]. Decision-level fusion is to make a final fusion decision based on the results of different algorithms; the methods include the voting method, the statistical method, and the fuzzy logic method [40,41]. The choice of the classification method has a crucial influence on the classification result. In recent years, the SVM classification method has been widely used in land cover classification research due to its advantages of requiring fewer samples and having high classification accuracy [42]. The biggest advantage of this method is that there is no need for data dimensionality reduction during classification and it shows high performance in terms of algorithm convergence, training speed, and classification accuracy.

The United Nations Convention to Combat Desertification and the Law of the People's Republic of China on Sand Prevention and Control were proposed to provide for land desertification prevention and desertification control. To evaluate the effectiveness of various ecological restoration projects, it is necessary to carry out dynamic monitoring and evaluation of sandy land and to grasp the current status and dynamic succession laws of sandy land in a timely and accurate manner. In this way, various policies and plans for desertification prevention and control can be implemented more efficiently [43,44]. Aiming at the scientific problem that the nature of sandy land is difficult to identify due to vegetation shading, the advantages of spectral characteristics of hyperspectral data and penetration characteristics of SAR data were made full use of for sandy land detection in this paper. HJ-2A data were used to carry out mixed-pixel decomposition in the horizontal direction to monitor the pure sand; SAR data were used to decompose and quantify the effect of vegetation covering on the sandy soil in the vertical direction in order to reveal the nature of the sandy land. The advantages of multi-source remote sensing data were fully used for sandy land detection, which provided technical support for dynamic monitoring of sandy land and prediction of desertification trend. The main objectives of this work are as follows: (i) with a focus on the advantages of multi-source data, proposing a new fusion method to detect sandy land, (ii) comparing the results of the new fusion method with those of others, such as mixed pixel decomposition and polarization decomposition, and (iii) determining the best and more accurate method to detect sandy land.

## 2. Study Area

The Otingdag Sandy Land in the Xilin Gol League of Inner Mongolia is located between 112°41′–117°30′ E and 42°06′–43°45′ N. This region has a mid-temperate continental climate. The annual rainfall is about 360 mm, mainly in July, August, and September. The Otingdag Sandy Land is one of the four major sandy lands in China, and most of its surface is covered by shifting, semi-fixed, and fixed sandy land. Its surrounding area is typical arid and semi-arid temperate grassland, and this area belongs to the ecotone of grassland and sandy land. Therefore, it is important to identify the boundaries of grassland and sandy land accurately and to monitor the range of the sandy land. This study took a partial area of the Otingdag Sandy Land as the study area, which is located at the junction of Sunite Left Banner, Abaga Banner, Zhengxiang White Banner, and Zhenglan Banner.

Alxa Left Banner is located in the west of Inner Mongolia. It is a temperate desert and arid area with a typical continental climate, characterized by heavy sandstorms, aridity, and strong evaporation. The annual rainfall is 80–220 mm, and the annual evaporation is 2900–3300 mm. Sandy lands cover an area of 34,000 km$^2$ and mainly include the Tengger Desert and the Ulanbuh Desert. This study took part of the area of Alxa as the verification area. The locations of the study area and the verification area are shown in Figure 1.

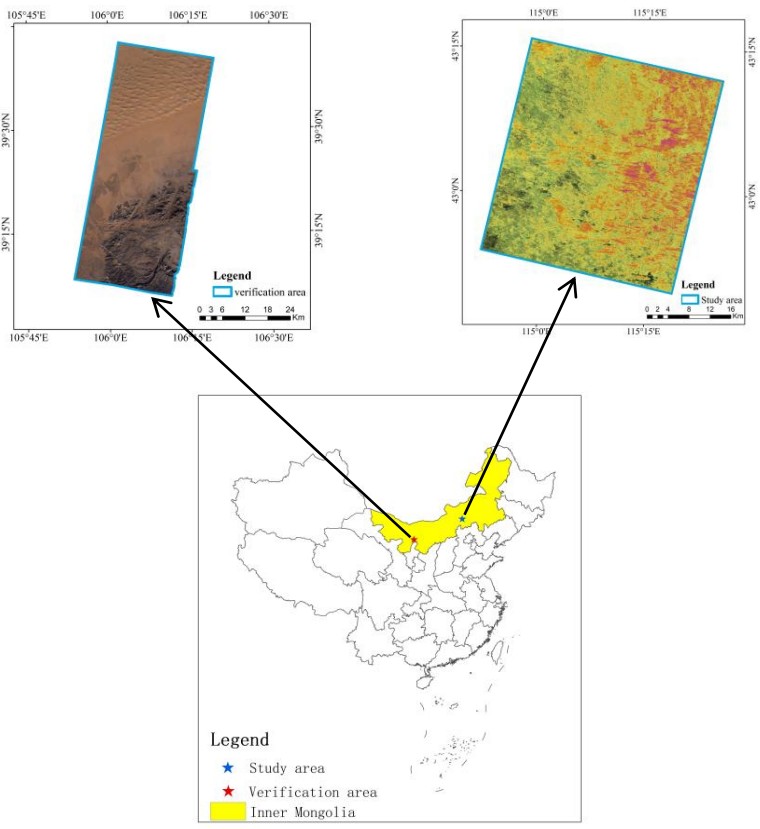

**Figure 1.** Locations of the study area and the verification area.

## 3. Data and Materials

### 3.1. Remote Sensing Data Acquisition and Processing

HJ-2A hyperspectral data, GF-3 SAR data, and Landsat 8 OLI multispectral data were gathered as the main remote sensing data. The specific image information is provided in Table 1. HJ-2A was launched by China on 27 September 2020. It is equipped with a wide-view CCD camera, an infrared multi-spectral scanner, and a hyperspectral imager. The HJ-2A hyperspectral data have a total of 215 bands, whose ability to detect and identify ground feature is greatly improved. Since the resolution of HJ-2A's visible light and near-infrared band is 48 m and the resolution of short-wave infrared is 98 m, this study used HJ-2A's high-resolution visible light and near-infrared bands, which have a total of 100 bands. GF-3 is China's first C-band radar satellite, with a high-resolution, fully polarized spaceborne SAR system. It has 12 conventional imaging modes, with a maximum spatial resolution of 1 m and a maximum width of 650 km. This study used full polarization strip 1 imaging mode data of GF-3, which provide more information than single-polarized SAR data and provide the possibility for the quantitative inversion of surface parameters. Landsat 8 OLI data contain nine different bands, whose imaging width can be up to 185 km. The panchromatic band range is narrow, which can better distinguish vegetation and non-vegetation areas. Therefore, it can be used to calculate vegetation coverage and provide reference information for accuracy assessment. The imaging time of HJ-2A and GF-3 were both in the non-growing season, in order to reduce the influence of vegetation on sandy land extraction. Landsat 8 OLI data were selected in the growing season to observe the vegetation cover.

**Table 1.** Remote sensing data information.

| Area | Type of Data | Imaging Time | Spatial Resolution | Image Quality |
|---|---|---|---|---|
| Otingdag | HJ-2A | 17 February 2021 | 48 m | No cloud coverage |
| | GF-3 | 5 January 2021 | 8 m | No cloud coverage |
| | Landsat 8 OLI | 13 August 2021 | 30 m | No cloud coverage |
| Alxa | HJ-2A | 2 February 2021 | 48 m | No cloud coverage |
| | GF-3 | 18 March 2020 | 8 m | No cloud coverage |
| | Landsat 8 OLI | 7 August 2021 | 30 m | No cloud coverage |

HJ-2A hyperspectral data were pre-processed by radiation calibration and atmospheric correction. Then, geometric correction was conducted based on Landsat 8 OLI image and the error was controlled to within 1 pixel. Firstly, GF-3 data were processed for multiple views to make the image's geometric features closer to the actual situation on the ground, while the speckle noise was reduced; secondly, it was processed for coherent speckle noise filtering; finally, geocoding was performed to re-sample and re-project the map.

Sensitive bands can be captured because hyperspectral remote sensing data have continuous spectral of ground features, which greatly improves the ability to detect sandy land. HJ-2A hyperspectral data were used to achieve mixed-pixel decomposition, which was based on the spectral information of remote sensing images to obtain sandy land information. SAR data have penetration characteristic in the vertical direction in order to reveal the nature of the sandy land. GF-3 data were used to achieve polarization decomposition, which can extract the main scattering features of ground objects effectively and realize sandy land detection accurately. Landsat 8 OLI data were used to conduct geometric correction for HJ-2A data. Meanwhile, they were used for vegetation coverage inversion to provide a judgment basis for the accuracy evaluation of the sandy land detection.

### 3.2. Field Data

According to the research tasks and goals, a research team was established and went to the research area. A total of 26 field samples were obtained through the research survey (Figure 2). The selection of the sample position was of vital importance; a sample position should be representative and include all land cover types and different types of vegetation coverage. GPS was used to obtain latitude and longitude information. The investigated content included land cover/land use, soil type, vegetation coverage, and sandy land degree, which were obtained to provide reliable verification data for the development of this study.

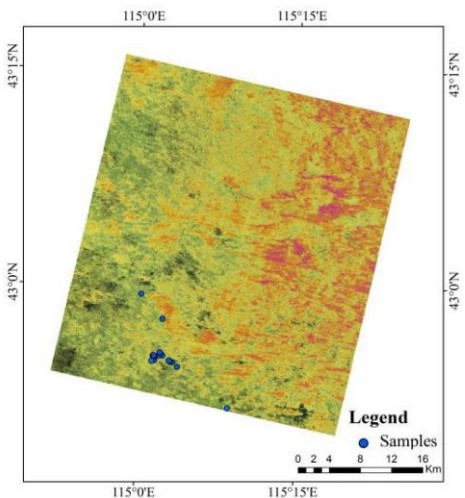

**Figure 2.** The distribution of field data.

## 4. Methodology

After pre-processing of multi-source remote sensing data, firstly, sandy land was detected based on a single HJ-2A hyperspectral image. The mixed-pixel decomposition method was used to obtain a sandy land abundance map and appropriate threshold selected for sandy land detection. Secondly, sandy land was detected based on GF-3 data. The polarization decomposition method was used to obtain polarization decomposition characteristics, and sandy land detection was realized by the support vector machine method. Then, sandy land was detected based on the fusion image of hyperspectral and SAR data; pixel-level fusion and feature-level fusion were used for sandy land detection, respectively. Finally, the field data were used to verify the accuracy of sand land detection. The technical flow chart is shown in Figure 3.

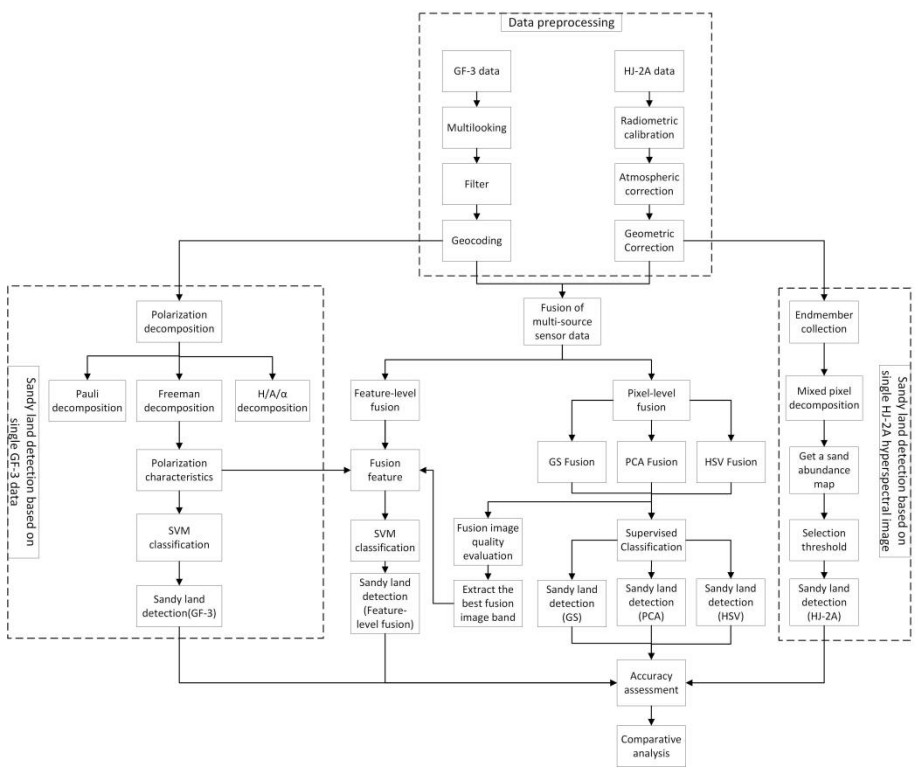

**Figure 3.** Technical flow chart.

### 4.1. Sandy Land Detection Based on a Single HJ-2A Hyperspectral Image

Firstly, the pure pixel index (PPI) was used to extract the pure pixels and the endmembers spectrum curve were obtained, which laid the foundation for the decomposition of mixed pixels. Secondly, the abundance layers of ground features in the remote sensing image were obtained based on the linear spectral unmixing (LSU) model. Finally, the sandy land could be detected by selecting an appropriate threshold.

Endmembers are the land cover types that make up a single pure spectrum of mixed pixels. The key step of mixed-pixel decomposition is endmember extraction [45], which directly affects the final decomposition accuracy. A random unit vector was mapped through N-dimensional scatter diagram iteration during the PPI process. The extremum pixels of each mapping were recorded, the DN value of each pixel indicating the number of times that the pixel was marked as an extremum. Therefore, the larger the pixel value, the higher the purity of the pixel.

The result of mixed-pixel decomposition was a series of grayscale images of the endmember, which indicated the proportion of the endmember spectrum in this pixel spectrum. The LSU model was formulated as follows (Equation (1)):

$$D_{Ni} = \sum_{j=1}^{p} m_{ij} \propto_j + e_i \tag{1}$$

where $i$ = 1, 2, ..., L; $j$ = 1,2, ..., $p$. Here, $i$ is the band index and $j$ is the endmember number, L is the total number of bands, and $p$ is the total number of endmembers; $m_{ij}$ represents the reflectance of endmember $j$ in band $i$; $\propto_j$ represents the proportion of the area endmember $j$ occupies in a pixel; and $e_i$ is the error in band $i$.

### 4.2. Sandy Land Detection Based on Single GF-3 Data

To reveal the main scattering characteristics of ground objects in GF-3 radar data, polarization characteristics caused by different scattering mechanisms of ground objects can be separated by polarization decomposition. Pre-processed GF-3 data were used to perform polarization decomposition by Pauli decomposition, H/A/$\alpha$ decomposition, and Freeman decomposition. Nine decomposition features were obtained as classification features, as shown in Table 2. Finally, sandy land detection was realized by the support vector machine method.

**Table 2.** Characteristic parameters corresponding to each polarization decomposition.

| Decomposition Method | Extract Features | Feature Meaning |
|---|---|---|
| Pauli Decomposition | $T_{11}$ | $T_{11}$ is the surface scattering information contained in Pauli decomposition. |
| | $T_{22}$ | $T_{22}$ is the dihedral scattering information contained in Pauli decomposition. |
| | $T_{33}$ | $T_{33}$ is the volume scattering information contained in Pauli decomposition. |
| H/A/$\alpha$ Decomposition | $\alpha$ | $\alpha$ is the average polarization scattering angle of H/A/$\alpha$ decomposition, identifying the main scattering mechanism. |
| | H | H is the polarization entropy of H/A/$\alpha$ decomposition, which measures the degree of polarization. |
| | A | A is the anisotropy of H/A/$\alpha$ decomposition, which measures the relative magnitude of non-dominant scattering. |
| Freeman Decomposition | $Odd_{F-D}$ | $Odd_{F-D}$ is the surface scattering power in Freeman decomposition. |
| | $Dbl_{F-D}$ | $Dbl_{F-D}$ is the dihedral scattering power in Freeman decomposition. |
| | $Vol_{F-D}$ | $Vol_{F-D}$ is the volume scattering power in Freeman decomposition. |

Pauli decomposition was established based on the polarization scattering matrix S, and each polarization basis matrix represented different types of ground objects [46]. The basic scattering matrix S can be expressed in Pauli basis as:

$$S = \begin{bmatrix} S_{HH} & S_{HV} \\ S_{VH} & S_{VV} \end{bmatrix} = \frac{a}{\sqrt{2}} \begin{bmatrix} 1 & 0 \\ 0 & 1 \end{bmatrix} + \frac{b}{\sqrt{2}} \begin{bmatrix} 1 & 0 \\ 0 & -1 \end{bmatrix} + \frac{c}{\sqrt{2}} \begin{bmatrix} 1 & 0 \\ 0 & 1 \end{bmatrix} + \frac{d}{\sqrt{2}} \begin{bmatrix} 1 & -j \\ j & 1 \end{bmatrix} \tag{2}$$

where $S_{HH}$ and $S_{VV}$ are co-polarized components; $S_{HV}$ and $S_{VH}$ are cross-polarized components; and a, b, c, and d are all complex numbers and represent the weights of the scattering matrix on 4 bases, respectively:

$$a = \frac{S_{HH} + S_{VV}}{\sqrt{2}}, \ b = \frac{S_{HH} - S_{VV}}{\sqrt{2}}, \ c = \frac{S_{HV} + S_{VH}}{\sqrt{2}}, \ d = j\frac{S_{HV} - S_{VH}}{\sqrt{2}} \tag{3}$$

The scattering matrix S was vectorized on the basis of Pauli decomposition to obtain the eigenvector K of the target as:

$$K = [a\ b\ c\ d] = \frac{1}{\sqrt{2}} [S_{HH} + S_{VV} \ \ S_{HH} - S_{VV} \ \ S_{HV} + S_{VH} \ \ i(S_{VH} - S_{HV})]^T \tag{4}$$

When the medium satisfies the mutually different condition, $S_{HV} = S_{VH}$, the Formula (4) becomes:

$$K = [a \, b \, c] = \frac{1}{\sqrt{2}}[S_{HH} + S_{VV} \quad S_{HH} - S_{VV} \quad 2S_{HV}]^{\,T} \tag{5}$$

The polarization scattering matrix S was reflected by Pauli coherent polarization decomposition to three basic scattering types, namely odd scattering, dihedral scattering with 0° rotation around the axis, and 45° dihedral scattering with rotation around the axis.

Freeman decomposition was the most representative of the decomposition methods based on the scattering model. It was divided into two-component and three-component decomposition. The three-component decomposition method was used to describe three basic scattering mechanisms by modeling, namely volume scattering, surface scattering, and dihedral angle scattering. Among them, volume scattering represents a group of small scattering objects with anisotropy, which represents vegetation and other ground object types. Secondary scattering refers to the scattering on two perpendicular scatterers, such as the commonly used radar corner reflector. Surface scattering refers to a medium roughness scatterer [45].

The covariance matrix corresponding to odd scattering was expressed as:

$$C_s = f_s \begin{bmatrix} |\beta|^2 & 0 & \beta \\ 0 & 0 & 0 \\ \beta^* & 0 & 1 \end{bmatrix} \tag{6}$$

The polarization covariance matrix corresponding to the second scattering was expressed as:

$$C_d = f_d \begin{bmatrix} |\alpha|^2 & 0 & \alpha \\ 0 & 0 & 0 \\ \alpha^* & 0 & 1 \end{bmatrix} \tag{7}$$

The polarization covariance matrix corresponding to volume scattering was expressed as:

$$C_v = f_v \begin{bmatrix} 1 & 0 & \frac{1}{3} \\ 0 & \frac{2}{3} & 0 \\ \frac{1}{3} & 0 & 1 \end{bmatrix} \tag{8}$$

where $f_s$, $f_d$, and $f_v$ are the surface, double-bounce, and volume (or canopy) scatter contributions to the VV cross section, respectively.

Three components of the Freeman decomposition were independent and irrelevant in the statistical. The sum of the three scattering mechanisms can be represented by the total covariance matrix obtained by the fully polarized SAR:

$$C = C_s + C_d + C_v \tag{9}$$

The total scattered power Span was expressed as:

$$\mathrm{Span} = C_{11} + C_{22} + C_{33} = f_s\left(1 + |\beta|^2\right) + f_d\left(1 + |\alpha|^2\right) + f_v \tag{10}$$

4.2.1. Sandy Land Detection Based on Pixel-Level Fusion

To detect sandy land accurately, three fusion methods, HSV, PCA, and GS, were adopted in this study (Table 3).

**Table 3.** Description of fusion methods.

| Fusion Methods | Fusion Effect |
| --- | --- |
| HSV Fusion | The edge information of the multi-spectral image, the target spectrum information, and the high-resolution features of the panchromatic image are retained. The texture details of the image are enhanced. |
| PCA Fusion | It has the function of data compression and information concentration. The information content of the first principal component is relatively high. When the panchromatic image is used to replace the first principal component for inverse transformation, the phenomenon of spectral distortion appears to a certain extent. |
| GS Fusion | The spectral information of the original multi-spectral image can be maintained, the spatial information is also significantly enhanced, and the spectral fidelity effect is better. |

#### 4.2.2. Sandy Land Detection Based on Feature-Level Fusion

The bands of the best fusion image and nine polarization features based on polarization decomposition were used as classification features to participate in the support vector machine classification. The mean value, standard deviation, entropy, and average gradient of the image were calculated as the indexes to evaluate the fusion image quality in this study.

The mean value referred to the average of pixel gray levels, which reflected the average brightness of the image. If the mean value was moderate, the visual effect was good.

$$\overline{M} = \frac{1}{m \times n} \sum_{x=1}^{m} \sum_{y=1}^{n} F(x, y) \tag{11}$$

where $m$ is the total number of rows of the image, $n$ is the total number of columns of the image, and $F(x,y)$ is the grayscale value of the $i$-th row and the $j$-th column of the image.

Entropy was an important indicator to measure the abundance of image information. Generally speaking, the greater the entropy, the more abundant the information contained in the image, and the better the fusion quality.

$$H(x) = -\sum_{i=0}^{255} p_i \log_2 p_i \tag{12}$$

where $p_i$ is the probability that the gray value of the image pixel is i.

The image's ability to express the contrast of small details can be sensitively reflected by the average gradient. So it can be used to evaluate the image clarity. Generally speaking, the larger the average gradient, the more the layers of the image, and the clearer the image.

$$g = \frac{1}{(m-1)(n-1)} \sum_{x=1}^{m-1} \sum_{y=1}^{n-1} \sqrt{\left(\left(\frac{\Delta F_x(x,y)}{\Delta x}\right)^2 + \left(\frac{\Delta F_y(x,y)}{\Delta y}\right)^2\right)/2} \tag{13}$$

where m is the total number of rows of the image, n is the total number of columns of the image, F(x,y) is the grayscale value of the i-th row and the j-th column of the image, $\frac{\Delta F_x(x,y)}{\Delta x}$ represents the gradient in the horizontal direction, and $\frac{\Delta F_y(x,y)}{\Delta y}$ represents the gradient in the vertical direction.

#### 4.3. Accuracy Verification Based on Field Data

The confusion matrix was adopted to analyze the quantitative accuracy evaluation. Image classification data and field samples were used for cross tabulation to provide multiple accuracy metrics, including overall accuracy, producer accuracy, and user accuracy. According to the field data, the sample point was judged to identify sandy land or non-sandy land in the results of sandy land detection and then it was compared with the

actual type of the sample point to determine whether the sample point was misclassified. Meanwhile, the user accuracy, the producer accuracy, and the overall accuracy of the sandy land detection were calculated.

## 5. Results

The vegetation coverage was used as auxiliary information to analyze the results of sandy land detection, as shown in Figure 4. Landsat 8 OLI data on August 13 was selected to invert the vegetation coverage in this study. During this period, the overall vegetation coverage of the sandy land was still at a relatively high level, and there was no artificial grassing operation.

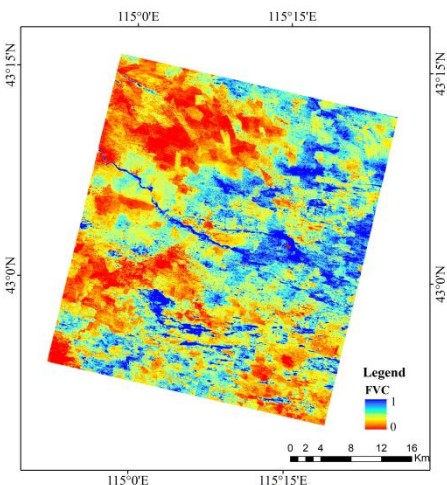

**Figure 4.** Distribution of vegetation coverage.

To ensure the scientific effectiveness of sandy land detection, different methods were used and verified for accuracy (Table 4).

**Table 4.** The accuracy of sandy land detection based on field data.

| Methods | User Accuracy | Producer Accuracy | Overall Accuracy |
| --- | --- | --- | --- |
| Decomposition of mixed pixels based on single-sensor HJ-2A | 63.64% | 82.35% | 60.00% |
| Polarization decomposition based on single-sensor GF-3 | 88.89% | 94.12% | 92.00% |
| Multi-source data GS pixel-level fusion | 68.00% | 100.00% | 72.00% |
| Multi-source data PCA pixel-level fusion | 60.87% | 82.35% | 56.00% |
| Multi-source data HSV pixel-level fusion | 60.87% | 82.35% | 56.00% |
| Multi-source data feature-level fusion | 88.89% | 94.12% | 92.00% |

### 5.1. Sandy Land Detection Based on a Single HJ-2A Hyperspectral Image

The pure pixel index (PPI) method was used to extract pure pixels. In the process of iterating pure pixels, the number of extracted pure pixels tended to be stable when the iterations number was 5000, which laid the foundation for the mixed-pixel decomposition. Finally, it was found that when the abundance of the sandy land endmember exceeded 50%, the pixel was definitely a sandy land type. Therefore, this threshold was used for sandy land detection (Figure 5).

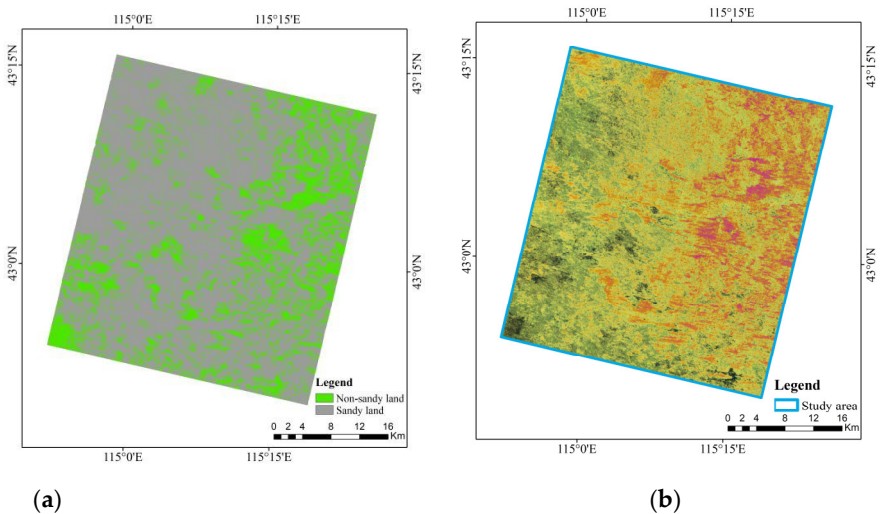

(**a**)  (**b**)

**Figure 5.** Sandy land detection based on an HJ-2A hyperspectral image. (**a**) Distribution of sandy land; (**b**) the image of HJ-2A hyperspectral.

The producer accuracy of sandy land detection based on the single-sensor HJ-2A hyperspectral image was 82.35%. However, the user accuracy was 63.64% and the overall accuracy was 60.00%. Combining sandy land detection and vegetation coverage, the identified sandy land areas were mainly distributed in the northwest of the image. The vegetation coverage in the northwestern region was sparse, and the detection effect of bare sand was good, but sandy land in the northeast areas with a high vegetation coverage was poorly detected. Therefore, a bare sand area could be well identified based on an HJ-2A hyperspectral image but sandy land covered by vegetation was difficult to identify.

### 5.2. Sandy Land Detection Based on Single GF-3 Data

Nine decomposition features were obtained by Pauli decomposition, H/A/α decomposition, and Freeman decomposition, and sandy land detection was realized by the support vector machine method (Figure 6).

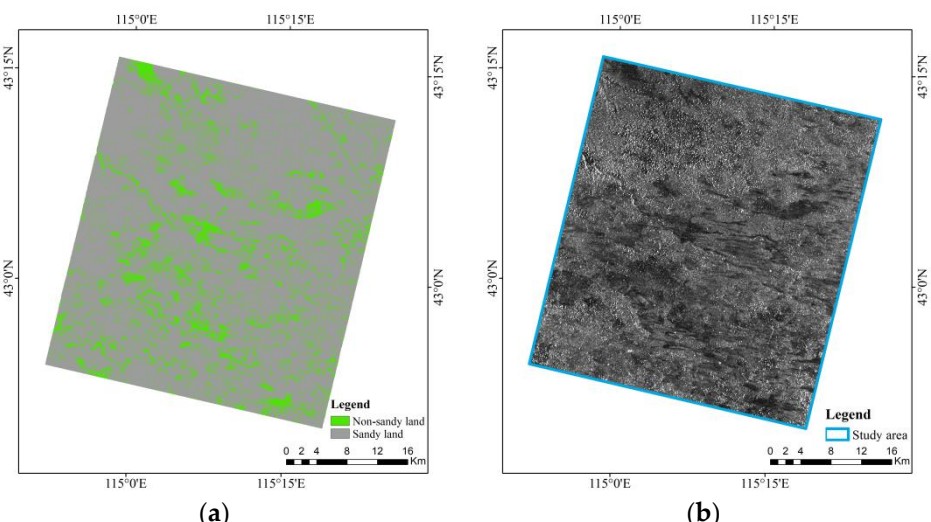

(**a**)  (**b**)

**Figure 6.** Sandy land detection based on GF-3 data. (**a**) Distribution of sandy land; (**b**) GF-3 data.

The user accuracy of sandy land detection based on single-sensor GF-3 data reached 88.89%, the producer accuracy was as high as 94.12%, and the total accuracy reached 92.00%. It can be seen that sandy land detection based on GF-3 data was considerable and the detected area of sandy land was relatively large, consistent with the actual situation. It can not only detect the exposed sandy land but also identify the nature of the sandy land with high vegetation coverage in the northeast. Compared with the accuracy of sandy land detection based on a single-sensor HJ-2 hyperspectral remote sensing image, the overall accuracy of sandy land detection based on a single-sensor GF-3 data was improved by 32%.

*5.3. Sandy Land Detection Based on a Fusion Image of HJ-2A and GF-3 Data*

5.3.1. Sandy Land Detection Based on Pixel-Level Fusion

The spectral curve of ground features was obtained by an HJ-2A image (Figure 7). The spectrum information of sandy land had an obvious peak inflection point at a wavelength of 570 nm; the spectrum curve of sandy land at 760 nm had an obvious valley inflection point, and the spectrum curve of sandy land was at the peak inflection point; three different types of land spectrum curves at 900 nm were at the peak inflection point and had no intersection. Because the wavelengths of 570 nm, 760 nm, and 900 nm were sensitive to sandy land information, GF-3 data and the corresponding wavelength band of HJ-2A were selected to perform GS fusion, HSV fusion, and PCA fusion, respectively. Then the appropriate training samples were selected based on the fusion image and the region of interest was calculated by the Jeffries–Matusita (JM) distance so that the conversion resolution obtained by the JM distance quantitatively was greater than 1.9 to ensure good separability between samples. Finally, the support vector machine classification method was used to obtain sandy land information (Figure 8).

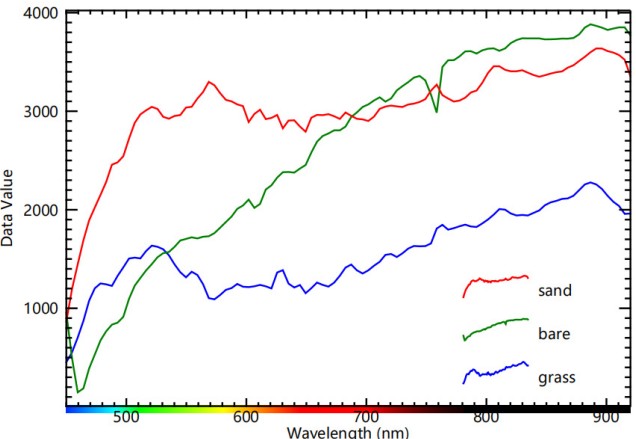

**Figure 7.** Spectral curves of different features.

The three methods based on pixel-level fusion showed relatively small differences in the distribution of sandy land. Compared with the method of sandy land detection from single HJ-2A data, the results of non-sandy land information detection near waters in the central and eastern regions were better. It can be seen from Table 4 that producer accuracy of the three fusion methods of sandy land detection was above 82%, indicating that the sandy sample points were correctly classified by these methods. However, the user accuracy and the overall accuracy were relatively low. Combined with the results of sandy land detection from pixel-level fusion images and the distribution of vegetation coverage, sandy areas with low vegetation coverage could be well identified but sandy areas with high vegetation coverage were not identified well.

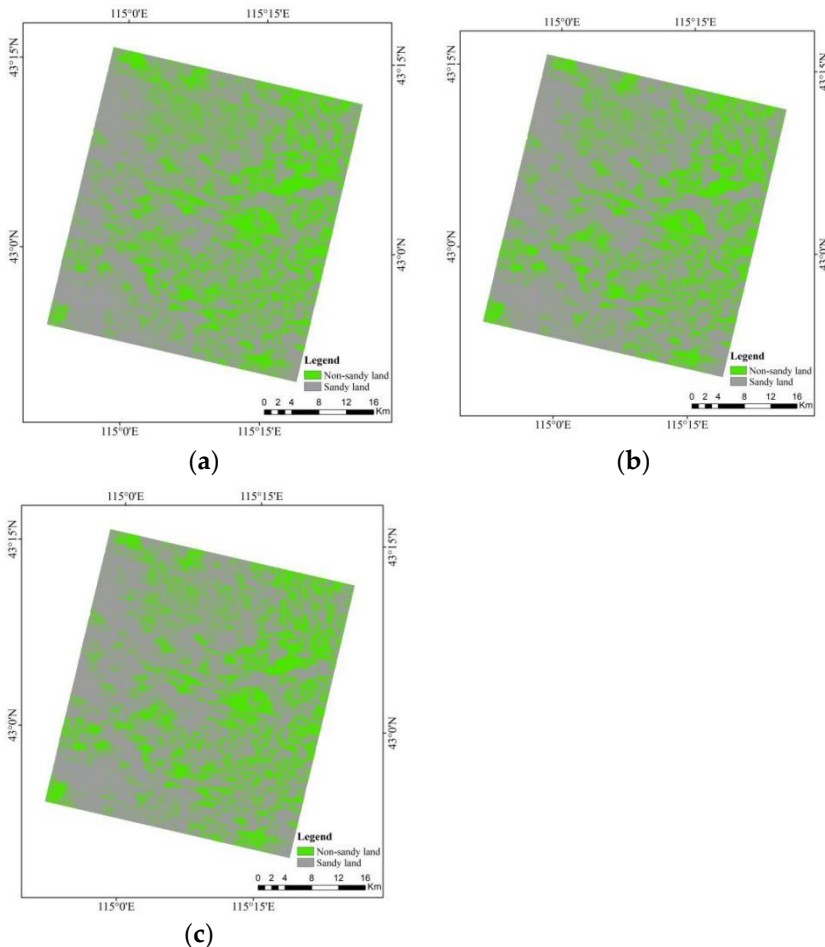

**Figure 8.** Sandy land detection based on pixel-level fusion. (**a**) Distribution of sandy land based on a GS fusion image; (**b**) distribution of sandy land based on a PCA fusion image; (**c**) distribution of sandy land based on an HSV fusion image.

5.3.2. Sandy Land Detection Based on Feature-Level Fusion

The mean value, standard deviation, entropy, and average gradient of the fusion results were calculated in this study so as to make a quantitative evaluation of the fusion images (Table 5).

**Table 5.** The quantitative evaluation indexes of multi-source data fusion.

| Methods | Mean | Standard Deviation | Entropy | Average Gradient |
|---------|------|--------------------|---------|--------------------|
| GS | 149.98 | 43.53 | 4.50 | 6.92 |
| PCA | 169.02 | 18.60 | 4.28 | 5.21 |
| HSV | 91.27 | 73.79 | 5.19 | 7.04 |

It can be seen from Table 5 that the mean value of HSV fusion was the lowest, that is, the image brightness was the lowest and had a certain degree of spectral distortion. The standard deviation of HSV fusion was the largest, which means that the gray levels in the HSV fusion image were scattered and the image quality was better. The entropy levels of three fusion images were relatively high, the entropy of the HSV fusion image reaching 5.19, indicating that the HSV fusion image had rich information and the best fusion quality. The average gradient of HSV fusion was the largest, which means that the HSV fusion image had a strong ability to express the contrast of small details and the image was clearer. To sum up, although the HSV fusion image had a certain degree of spectral distortion, the information it provided was rich and the image details were clearer.

Therefore, the bands of the HSV fusion image and nine polarization features based on polarization decomposition were used as classification features for support vector machine classification. After many experiments, it was found that the best classification result was obtained when the polynomial kernel function was selected and the degree was set to 2 using PolSARpro software to achieve SVM classification (Figure 9).

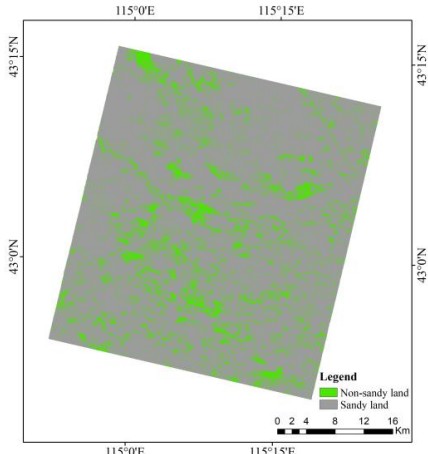

**Figure 9.** Sandy land detection based on feature-level fusion.

The user accuracy of sandy land detection based on the feature-level fusion reached 88.89%, the producer accuracy was as high as 94.12%, and the overall accuracy was 92.00%. Combined with vegetation coverage, it was found that the results of sandy land detection included bare sandy land with low vegetation coverage in the northwest and the sandy land with high vegetation coverage in the northeast. Sandy land's nature can be well recognized by fusing the hyperspectral and radar data, and the detection accuracy was improved a lot because not only the problem of mixed pixels in the sparse-vegetation scene in the horizontal direction was solved but the problem of the sandy land covered by vegetation in the vertical direction can also be solved.

*5.4. The Accuracy Evaluation of Sandy Land Detection Based on a Google Earth Image*

The field data had 26 sample points and were unevenly distributed in the study area (Figure 2). To ensure the scientific validity of sandy land detection, a grid was constructed for the study area so that the sample points were evenly distributed in space (Figure 10). The high spatial resolution of Google Earth images was used to identify the topographic categories of sample points, which verified the accuracy of sandy land detection by different methods (Table 6).

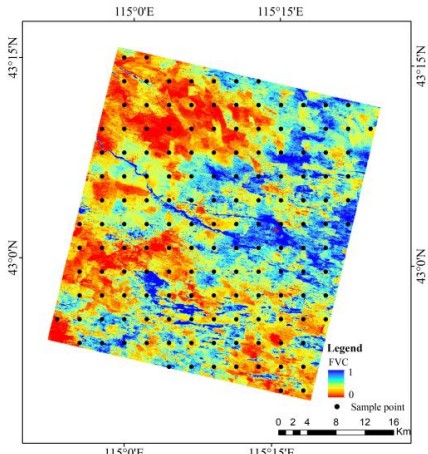

**Figure 10.** Distribution of sample points in the grid of a Google Earth image.

**Table 6.** The accuracy of sandy land detection based on a Google Earth image.

| Methods | User Accuracy | Producer Accuracy | Overall Accuracy |
|---|---|---|---|
| Decomposition of mixed pixels based on single-sensor HJ-2A | 94.34% | 75.19% | 74.17% |
| Polarization decomposition based on single-sensor GF-3 | 93.89% | 92.48% | 88.08% |
| Multi-source data GS pixel-level fusion | 98.02% | 74.44% | 76.16% |
| Multi-source data PCA pixel-level fusion | 97.12% | 75.94% | 76.82% |
| Multi-source data HSV pixel-level fusion | 96.36% | 79.70% | 79.47% |
| Multi-source data feature-level fusion | 93.94% | 93.23% | 88.74% |

It can be seen from Table 6 that the user accuracy of sandy land detection based on a single-sensor HJ-2A hyperspectral image was as high as 94.34% but its producer accuracy and overall accuracy were just 75.19% and 74.17%, respectively, which were relatively lower. According to the distribution of sandy land, the result was good in identifying bare sandy areas but the sandy land covered by vegetation was difficult to identify. The user accuracy of sandy land detection based on single-sensor GF-3 data was as high as 93.89%; meanwhile, the producer accuracy was as high as 92.48% and the total accuracy was 88.08%. Not only can bare sandy land be detected, but the nature of the sandy land in areas with high vegetation coverage in the northeast can also be identified. Compared with the result of sandy land detection based on a single hyperspectral image, the overall accuracy was improved by 14%. The user accuracy of sandy land detection based on the feature-level fusion was as high as 93.94%, the producer accuracy was as high as 92.23%, and the total accuracy was 88.74%, which were slightly better than sandy land detection based on single GF-3 data. Therefore, accurate identification of sandy land can be realized based on the feature-level fusion.

### 5.5. Application Promotion and Verification Analysis

The HJ-2A hyperspectral data were a new sensor. The data that can be obtained were limited, and the fully polarized GF-3 data were less in the Otingdag Sandy Land. The above reasons led to a small study area covered and an imbalance in the number of sample points of different land types.

At the same time, the overall accuracy of sandy land detection based on feature-level fusion was slightly higher than that based on single-sensor GF-3, though there may be some discrepancy. Therefore, in order to prove the feasibility and universality of the method of sandy land detection based on feature-level fusion, part of the area of Alxa was selected as the verification area. The distribution of vegetation coverage and sample points are shown in Figure 11.

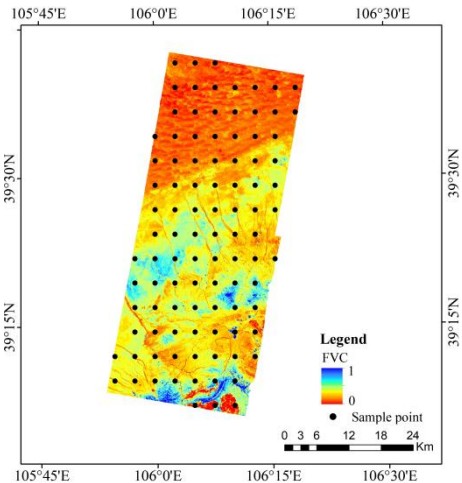

**Figure 11.** Distribution of vegetation coverage and sample points.

Combined with the distribution of vegetation coverage (Figure 11) and the sandy land detection (Figure 12), the sandy land was mainly distributed in the northern part of the verification area. The intermediate transition zone was mostly sandy land covered by sparse vegetation. However, there were also small sand belts passing by here, which were clearly separated from the non-sandy land. It can be seen from Table 7 that the user accuracy of the multi-source data feature-level fusion was as high as 95.65%, the producer accuracy was 95.65%, and the total accuracy was 94.90%, which were higher than the accuracy obtained by using the single-sensor GF-3 to detect sandy land. In addition, the nature of sandy land covered by sparse vegetation in the middle of verification area can be identified well.

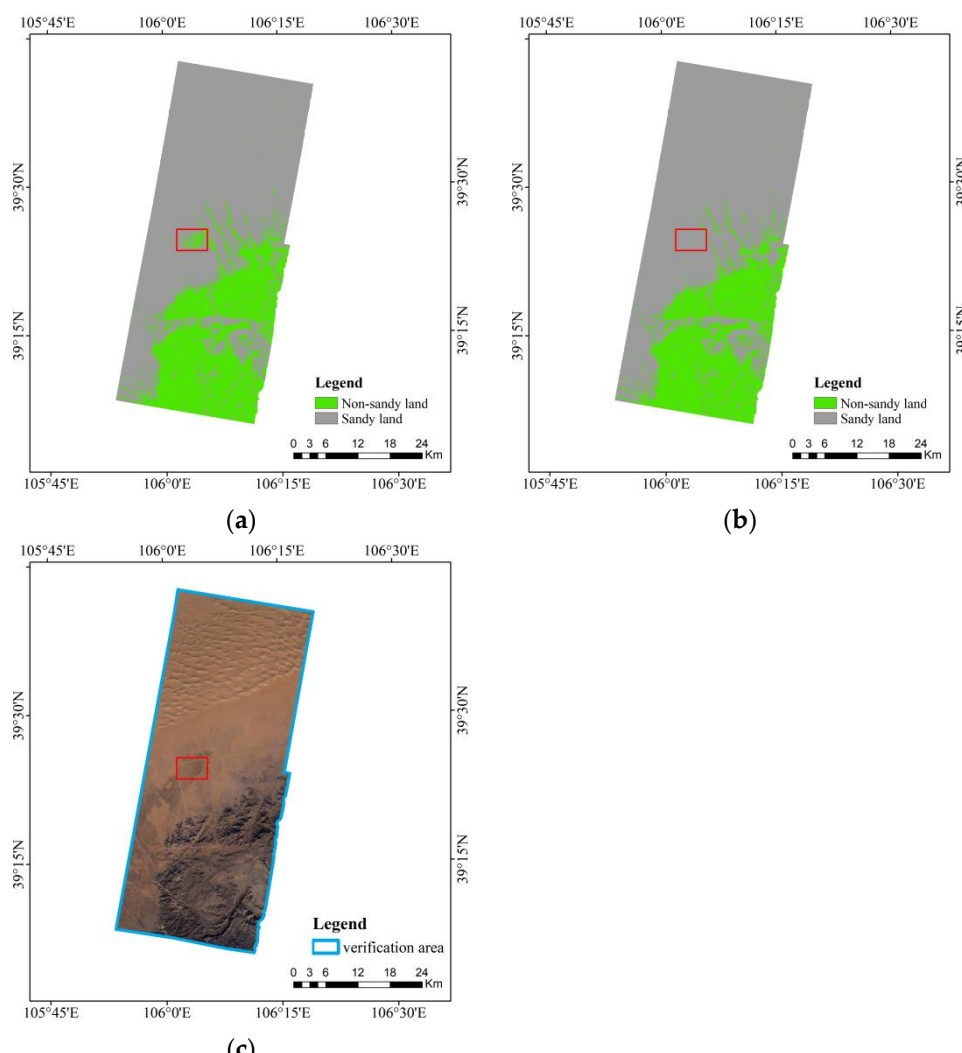

**Figure 12.** Sandy land detection in the verification area. (**a**) Sandy land detection based on single-sensor GF-3; (**b**) sandy land detection based on feature-level fusion; (**c**) HJ-2A hyperspectral image in the verification area.

**Table 7.** The accuracy of sandy land detection in the verification area.

| Methods | User Accuracy | Producer Accuracy | Overall Accuracy |
|---|---|---|---|
| Polarization decomposition based on single-sensor GF-3 | 94.12% | 92.75% | 91.80% |
| Multi-source data feature-level fusion | 95.65% | 95.65% | 94.90% |

## 6. Discussion

The classification result can be improved by the fusion image of hyperspectral images and SAR data, which can not only highlight the hyperspectral thematic information and eliminate or suppress irrelevant information but also retain the backscatter information of the target in the SAR image and detect the vertical structure information of the ground object.

According to the sandy land detection results in this study, sandy land with low vegetation coverage can be identified by different methods, that is, the nature of bare sandy land can be better revealed. According to the accuracy of sandy land detection based on field data, it can be seen that sandy land detection based on a single hyperspectral image and pixel-level fusion was poor and the nature of sandy land covered by vegetation could not be identified well. The results of sandy land detection based on single GF-3 data and feature-level fusion had the same accuracy. Not only can better information on bare sand in the horizontal direction be acquired, but the problem of information on sand covered by sparse vegetation in the vertical direction can also be better resolved. According to the accuracy of sandy land detection based on Google Earth images, it can be seen that when the number of sample points was large and the points evenly distributed, the verification accuracy of sandy land detection was improved. Compared with the method of sandy land detection with single-sensor data, the accuracy of detecting bare sandy land based on pixel-level fusion was improved but the nature of sandy land covered by vegetation was still difficult to identify. Sandy land detection based on feature-level fusion was slightly better than that based on single GF-3 data in terms of user accuracy, producer accuracy, and overall accuracy. Not only can bare sandy land be accurately detected, but the nature of sandy land covered by vegetation can also be revealed through decomposing and quantifying the impact of vegetation on soil.

According to sandy land detection in the verification area, it can be seen that sandy land detection based on feature-level fusion was better than that based on single GF-3 data in terms of user accuracy, producer accuracy, and overall accuracy and the nature of the sandy land covered by sparse vegetation in the middle of verification area can be better identified. Due to the large area of verification data and the balanced number of samples of different land types, the results were more objective and the methods were more feasible and repeatable. Therefore, it can be concluded that not only can bare sand be identified better but the nature of the sand covered by sparse vegetation can also be identified.

The Normalized Differential Sandy Areas Index (NDSAI) was proposed by Sahar et al. to separate the sandy areas from the non-sandy areas. Because SWIR1 is highly sensitive to vegetation, it was used to develop the new NDSAI. Therefore, NDSAI can better distinguish sandy land from vegetation [47]. The feature-level fusion algorithm proposed in this article focuses on the advantage of data to reveal the nature of sandy land in sparse-vegetation scenes. Its accuracy was slightly better than that of NDSAI, but the data processing was more complicated. Tan et al. proposed an unsupervised classification method based on fully polarimetric SAR data. It integrated three key steps: new decomposition (ND), super pixel segmentation, and LSC algorithm. The polarization parameters can be obtained effectively by ND-LSC, and the classification accuracy was improved [48]. SAR data were also used in this article, focusing more on SAR's penetration characteristic in the vertical direction. Multi-source data fusion and the SVM classification method were used to detect sandy land effectively.

In summary, the advantages of hyperspectral and SAR data were fully used by the feature-level fusion. The problem of mixed pixels in the sparse-vegetation scene in the horizontal direction was solved, as was the problem of vegetation covering the sandy land in the vertical direction, which provided favorable technical support for sandy land detection.

## 7. Conclusions and Prospects

In this study, an HJ-2A hyperspectral image was used for mixed-pixel decomposition to detect sandy land horizontally and GF-3 data were used for polarization decomposition to detect sandy land vertically, while the fusion image of HJ-2A and GF-3 data was used to detect sandy land horizontally and vertically. The results of sandy land detection were evaluated for accuracy based on the distribution of vegetation coverage, field data, and sample points based on Google Maps. The results showed that when the exposed sandy land is widely distributed in the study area, the pixel-level fusion based on GS can be used to detect sandy land. When the vegetation coverage is high and the sandy land is mostly covered by vegetation, feature-level fusion is a good selection to be used to detect sandy land. Based on the feature-level fusion method, the nature of the sandy land covered by vegetation is revealed and accurate identification of sandy land can be realized. The method proposed in this paper made full use of the horizontal and vertical structure information of remote sensing data. The problem of mixed pixels in the sparse-vegetation scene in the horizontal direction was solved, as was the problem of vegetation covering sandy land in the vertical direction. It can provide an important reference for sandy land detection in arid and semi-arid areas and provide technical support for sandy land prevention and control. In the follow-up study, wide-range and high-resolution data will be used to verify the method of sandy land detection based on the feature-level fusion image of hyperspectral and radar data proposed in this paper. In addition, a new fusion method will be developed to improve the fusion accuracy in order to further improve the accuracy of sandy land detection.

**Author Contributions:** Conceptualization, J.W.; methodology, J.W. and Y.L.; software, J.W., X.D. and B.S.; validation, J.W., Y.L. and Q.L.; formal analysis, J.W. and Y.L.; investigation, J.W., Y.L. and A.Y.; resources, J.W. and Y.L.; data curation, Y.L., K.X., K.A. and X.W.; writing—original draft preparation, Y.L.; writing—review and editing, J.W.; visualization, J.W., Y.L., F.C. and M.S.; supervision, J.W., X.S. and B.Z.; project administration, J.W.; funding acquisition, J.W. All authors have read and agreed to the published version of the manuscript.

**Funding:** This research was funded by the National Natural Science Foundation of China's Youth Science Foundation Project (project nos. 42001360 and 42001386).

**Institutional Review Board Statement:** Not applicable.

**Informed Consent Statement:** Not applicable.

**Data Availability Statement:** The image data of HJ-2A, GF-3, and Landsat-8 satellites were downloaded from the official websites http://36.112.130.153:7777/#/home/ (accessed on 20 April 2021), http://ids.ceode.ac.cn (accessed on 7 May 2021), and https://earthexplorer.usgs.gov/ (accessed on 22 September 2021).

**Conflicts of Interest:** The authors declare no conflict of interest.

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
