# Peer review of "Methods of Sandy Land Detection in a Sparse-Vegetation Scene Based on the Fusion of HJ-2A Hyperspectral and GF-3 SAR Data"

_remotesensing, doi:10.3390/rs14051203_

Round 1

Reviewer 1 Report

This article is based on a rigorous scientific approach both in methodology and in the analysis of results in line with the publication of many works of the same category. It is also in line with articles published by the same team (i.e. Geoderma, 316, 2018, 89-99).

To be noted: it could be interesting that the authors integrate recent publications and results in their analyze as:

  • SAHAR, Awad A. et al.(2021)-  Mapping Sandy Areas and their changes using remote sensing. A Case Study at North-East Al-Muthanna Province, South of Iraq. Revista de Teledetección, n. 58, p. 39-52, july 2021 Date accessed: 02 feb. 2022. doi.org/10.4995/raet.2021.13622.
    - WEIXIAN Tan et al. (2021) - A Novel Unsupervised Classification Method for Sandy Land Using Fully Polarimetric SAR Data. Remote Sens. 2021, 13, 355. doi.org/10.3390/rs13030355
  • However, can real scientific added value be achieved only by multiplying the papers by changing the sensors and/or the geographical areas? The reviewer's preference is clearly towards methodological comparison papers (benchmarking) or innovative approaches (AI-DL) that are most likely to be effective in the field addressed by this paper.

  • In conclusion, even if the article is nearly publishable as is, the reviewer strongly encourages the authors to include it in a methodological synthesis that could be a landmark rather than producing an article of limited interest. The authors team has the talent and potential...

Reviewer 2 Report

This paper examined an integrated method for SAR and optical images for land surface classification, which looks very useful and can be considered for publication after some modifications about the description of the classification technique.

In Figure 3, the authors used SVM classification for the sandy land detection from the fusion of SAR and optical images. However, a sufficient explanation about why you chose SVM from a lot of classification methods, particularly many non-linear AI classification methods. Also, how were the hyperparameters for the SVM such as Kernel, C, and Gamma set up for a more optimized classification?

Reviewer 3 Report

In their work, the authors fused HJ-2A hyperspectral and GF-3 SAR data in order to improve sandy land detection. The results obtained confirm the effectiveness of the detection approach proposed by the authors.

In general, the study is interesting but needs to be improved.

In the abstract and in the introduction a novelty of the method presented by the authors in comparison with those described in the literature should be formulated. The authors should more clearly list what is significant done in this study.

The authors indicate that the main data sources were HJ-2A and GF-3 data, however later in the paper Landsat 8 OLI data is also used. The difference in approach to the use of these data should be explained.

On Fig. 1, the Legend font is too small, hard to read (e.g. study and verification area).

Section 3.2. A more detailed description of field samples is required.

Figure 3. Probably, PCA fusion (not PC fusion).

The description of the expressions cannot be considered satisfactory. In a number of places there is no description of variables.

For example, in (1) it is indicated that j=1…P, however the notation p is used in the equation. The meaning of L, mij, etc. is not explained.

The text uses designations such as SVV, SVH, etc. and in the equations SVV,  SVH,  etc. I would recommend bring them to the same format.

(6)-(8) - it should be explained what fs, fd, and fv are.

(11)-(13) are also given without explanation.

Round 2

Reviewer 3 Report

The authors did the necessary work and provided answers to my questions and comments. I think that the article can be published in this version.

There is a small note - in the equation (1) there is a lowercase p, and in the following description - an uppercase P.

Author Response

This manuscript is a resubmission of an earlier submission. The following is a list of the peer review reports and author responses from that submission.